# The Potentiality of Natural Products and Herbal Medicine as Novel Medications for Parkinson’s Disease: A Promising Therapeutic Approach

**DOI:** 10.3390/ijms25021071

**Published:** 2024-01-15

**Authors:** Yu-Jin So, Jae-Ung Lee, Ga-Seung Yang, Gabsik Yang, Sung-Wook Kim, Jun-Ho Lee, Jong-Uk Kim

**Affiliations:** 1College of Korean Medicine, Woosuk University, Jeonju-si 54986, Jeollabuk-do, Republic of Korea; ujso4334@naver.com (Y.-J.S.); jwlee4532@naver.com (J.-U.L.); didrktmd@naver.com (G.-S.Y.); yanggs@woosuk.ac.kr (G.Y.); sheep.sw91@gmail.com (S.-W.K.); 2Da CaPo Co., Ltd., 303 Cheonjam-ro, Wansan-gu, Jeonju-si 55069, Jeollabuk-do, Republic of Korea

**Keywords:** Parkinson’s disease, natural products, alpha synuclein, neuroinflammation, oxidative stress, mitochondrial dysfunction, neurodegeneration

## Abstract

As the global population ages, the prevalence of Parkinson’s disease (PD) is steadily on the rise. PD demonstrates chronic and progressive characteristics, and many cases can transition into dementia. This increases societal and economic burdens, emphasizing the need to find effective treatments. Among the widely recognized causes of PD is the abnormal accumulation of proteins, and autophagy dysfunction accelerates this accumulation. The resultant Lewy bodies are also commonly found in Alzheimer’s disease patients, suggesting an increased potential for the onset of dementia. Additionally, the production of free radicals due to mitochondrial dysfunction contributes to neuronal damage and degeneration. The activation of astrocytes and the M1 phenotype of microglia promote damage to dopamine neurons. The drugs currently used for PD only delay the clinical progression and exacerbation of the disease without targeting its root cause, and come with various side effects. Thus, there is a demand for treatments with fewer side effects, with much potential offered by natural products. In this study, we reviewed a total of 14 articles related to herbal medicines and natural products and investigated their relevance to possible PD treatment. The results showed that the reviewed herbal medicines and natural products are effective against lysosomal disorder, mitochondrial dysfunction, and inflammation, key mechanisms underlying PD. Therefore, natural products and herbal medicines can reduce neurotoxicity and might improve both motor and non-motor symptoms associated with PD. Furthermore, these products, with their multi-target effects, enhance bioavailability, inhibit antibiotic resistance, and might additionally eliminate side effects, making them good alternative therapies for PD treatment.

## 1. Introduction

Parkinson’s Disease (PD) is the second most common neurodegenerative disease worldwide [1]. The incidence rate of PD in women shows an increasing trend from 2.94 per 100,000 persons in the age group 40–49 years to a peak of 104.99 per 100,000 between 70–79 years, which then declines to 66.02 per 100,000 in those aged 80 and above. For men, it rises from 3.59 per 100,000 in the 40–49 years age group to a peak of 132.72 per 100,000 between ages 70 and 79, and then drops to 110.48 per 100,000 in those 80 and older. Both genders display an increased incidence rate with age [2]. The prevalence of PD in Asia is lower compared to the West. According to the age-standardized 2000 WHO population, the prevalence of PD in Asian studies was found to be 51.3–176.9/100,000 from field surveys and 35.8–68.3/100,000 from record-based surveys, whereas Western studies reported higher prevalence rates of 101–439.4/100,000 from field surveys and 61.4–141.1/100,000 from record-based surveys [3]. The number of workdays lost due to PD increased from 27.6 days three years before diagnosis to 61.5 days three years after diagnosis, and indirect costs increased from $3549 to $7892. Caregivers also took more time off work after the diagnosis, resulting in an indirect cost rising from $302 to $850 [4]. With increasing longevity, the prevalence of PD is expected to double within the next 20 years. Without more effective treatments, the social and economic burdens associated with PD are anticipated to grow [5].

PD typically manifests with symptoms that gradually develop over several years, making diagnosis difficult [6]. On average, 15 years after PD onset, patients frequently experience falls, become immobile, show cognitive decline, and are often admitted to care facilities in an exacerbated state [7]. Additionally, out of 178 idiopathic PD patients, 109 (61.2%) died during the tracking period. Among the deceased, 77 (53.8% of 143) were from the PD group, 12 (92.3% of 13) were from the multiple system atrophy group, and 16 (88.9% of 18) were from the progressive supranuclear palsy group [8]. PD patients had a higher mortality rate compared to controls across all age groups and genders. Other studies also indicate that PD is a significant factor in increased mortality, regardless of age and gender [9,10,11].

Symptoms of PD can be divided into non-motor neurological abnormalities and motor abnormalities. Non-motor symptoms include olfactory dysfunction, cognitive impairment, psychosis, and sleep disorders [12,13]. Motor symptoms manifest as tremors, rigidity, and gait abnormalities [14]. PD progresses in three stages: the preclinical phase, where neurodegenerative processes start but do not yet cause noticeable symptoms; the prodromal phase, where noticeable symptoms arise from the evident neurodegeneration in various regions of the central nervous system (CNS) and peripheral nervous system (PNS); and the clinical phase, where classical motor symptoms become prominently evident [15].

Another issue with PD is the emergence of complications. Postural instability and associated falls, combined with other central features, are among the most common complications in PD patients [16,17]. Other issues like tremors, abnormal movements, dementia, depression, hallucinations, and autonomic nervous system dysfunction symptoms are also observed in PD patients [18].

PD is difficult to control with medical treatment, especially with the continuous growth in the elderly population [19]. Although drugs have been developed to delay PD progression, current medications have limitations and side effects [20,21,22]. Therefore, with the limitations and side effects of commonly used drugs, there is a demand for new medicines without these drawbacks. Consequently, there has been a surge in research on traditional Korean medicine and natural extracts [23,24,25]. Therefore, this study aims to investigate the potential of herbal medicines and plant-derived extracts used in traditional Korean medicine for improving PD.

## 2. Pathophysiology of Parkinson’s Disease

PD arises from the interplay of various genetic and environmental factors, with the primary pathology being the abnormal accumulation of the α-synuclein (α-Syn) protein within neurons. This irregular accumulation impacts the dopamine-producing neurons in the substantia nigra, eventually leading to neuronal death. This results in an imbalance of dopamine levels, inducing various motor symptoms of PD [26,27,28]. Additionally, pathological observations in PD include mitochondrial dysfunction, lysosomal disorders, and inflammation (Figure 1).

### 2.1. The Main Cause of Parkinson’s Disease

Beyond the loss of dopaminergic neurons, a pathological diagnosis of PD requires the presence of Lewy bodies, consisting of protein α-Syn, in the neurons [29]. α-Syn, comprising 140 amino acids, is a neuron-specific protein abundantly found in synapses and potentially participates in neurotransmission [30]. At a neutral pH, α-Syn exists in an irregular form, but it is assumed to form an α-helix structure when bound to membranes or vesicles containing acidic lipids. In its normal state, α-Syn is believed to assist synaptic vesicle release, providing a stabilizing effect [31]. However, in PD, mutations in the α-Syn-related gene or post-translational modifications such as cleavage, nitrosylation, ubiquitination, and phosphorylation, transform a-Syn into an aggregation-prone toxic beta-sheet form [32]. Many studies support the notion that small amounts of misfolded pathological α-Syn can accelerate the misfolding of endogenous α-Syn, acting as a template and amplifying the pathology [33,34]. Lewy bodies are pathological products primarily composed of α-Syn, but also incorporate hundreds of other proteins. Recent studies suggest a relationship between Lewy bodies and aggresomes [35]. The misfolding and pathological changes of α-Syn differ depending on conditions. In multiple system atrophy, the cellular environment of oligodendrocytes has been observed to intensify the altered forms of α-Syn, leading to increased toxicity [36]. α-Syn exhibits prion-like characteristics. Abnormal α-Syn is released from donor neurons and endocytosed into recipient neurons, propagating from one neuron to another [37]. α-Syn aggregates can be transported in both anterograde and retrograde directions along axons. Notably, synaptic contact is not necessarily required for transfer from axons to neuronal bodies [38]. Neuronal degeneration can occur without the spread of α-Syn pathology and the formation of Lewy bodies [39]. This is supported by clinical trial results wherein PD patients who underwent dopaminergic neuron transplantation showed motor function improvement for up to 18 years after transplantation without pharmacological dopamine treatment [40].

According to the Braak’s staging model, PD begins its development associated with the olfactory brain or the peripheral autonomic system and adrenal medulla [41,42]. It then progresses through the brainstem, followed by cerebral pathways, higher cortical areas responsible for autonomous function regulation, higher-order sensory association regions, frontal areas, and eventually the entire neocortex (from stages 1 to 6). Based on this model, pathological changes in Lewy bodies are observed in the substantia nigra by stage 3 [43], and stages prior to this are considered to be asymptomatic or pre-symptomatic. Non-motor symptoms, such as olfactory and autonomic symptoms (e.g., gastrointestinal and urinary symptoms), appear during the pre-symptomatic stage [44]. However, some retrospective clinical–pathological studies have shown that at least 15% of PD patients did not have α-Syn progression in accordance with Braak’s criteria [45]. Pathological changes can initially start in areas like the brain’s relay nuclei or the olfactory brain and propagate downward to the brainstem [46].

Furthermore, recent studies have evidenced that α-Syn pathology can propagate from the enteric nervous system to the brain [47]. Several pathways, including the vagal nerve route, immune system mediators, gut-related hormones, and microbial-derived signaling molecules, have been suggested, but further research is required to identify the exact mechanism of the transmission of α-Syn pathology from the gut to the brain [48]. Neuroimaging techniques such as positron emission tomography and magnetic resonance imaging (MRI) have evaluated structural damages at various levels in the peripheral autonomic system and CNS. In MRI studies, structural changes in the brainstem, including prominent pontine lesions, were observed in body-predominant PD patients [49,50]. The detection rates of α-Syn in the skin and intestines were also higher in body-predominant PD compared to brain-predominant PD [51,52,53,54,55]. Dopamine transporter imaging revealed more asymmetric degeneration of the nigrostriatal pathway in brain-predominant PD [56].

### 2.2. Mitochondrial Dysfunction

Mitochondria exhibit a decline in quality and function with aging, and mitochondrial dysfunction has been observed in various neurodegenerative diseases [57]. Mitochondrial dysfunction has been recognized as a significant factor in the loss of dopaminergic neurons [58].

The first hint of a crucial role of mitochondria in the pathogenesis of PD was found in the 1980s when a group of heroin users in California developed PD induced by 1-methyl-4-phenyl-1,2,3,6-tetrahydrodropyridine (MPTP) [59]. Various toxins, such as MPTP, rotenone, paraquat, pyridaben, trichloroethylene, and fenpyroximate, have been associated with PD-like symptoms, have been shown to inhibit mitochondrial complexes [60], and the resulting disturbances in energy production and cellular functions could lead to damage in dopaminergic neurons and symptoms similar to PD.

Patients with PD exhibit reduced Complex I activity in the CNS and the prefrontal cortex [61,62]. This is due to the detrimental impact of abnormal α-Syn expression on mitochondrial function [26,63]. The N-terminus of α-Syn possesses a mitochondrial inner membrane-targeting sequence, and it interacts with Complex I, resulting in reduced activity both in vitro and in vivo. Additionally, mitophagy, a type of macroautophagy that selectively removes dysfunctional mitochondria to maintain mitochondrial homeostasis, is also associated with PD. Incomplete mitophagy, resulting from mutations in the genes Parkin and PTEN-induced putative kinase 1 (PINK1), causes autosomal recessive PD and is also associated with idiopathic PD [64].

A few genes have been confirmed to cause familial PD [65,66], and they are directly associated with autosomal dominant mutations in Synuclein Alpha (SNCA) and Leucine-rich repeat serine/threonine-protein kinase 2 (LRRK2) and autosomal recessive mutations in Parkin and PINK1 [67]. Parkin promotes mitophagy by attaching ubiquitin to outer mitochondrial membrane proteins [68]. However, Parkin mutations lead to increased p53 expression, which can potentially induce cell death [69]. Located in the mitochondria, PINK1 protects neuronal cells during acute oxidative stress, and promotes mitophagy. Defects in PINK1 lead to mitochondrial dysfunction and accumulation. As PD patients age, PINK1 plays a role in the loss of dopaminergic neurons [70]. DJ-1 is a multifunctional protein involved in transcriptional regulation, mitochondrial protection, antioxidant activities, and chaperone activation [71]. Being essential for mitophagy, defects in DJ-1 result in incomplete mitophagy, contributing to autosomal recessive PD [72]. LRRK2 is found in the mitochondrial outer membrane, nucleus, and cytoplasmic membrane. Mutations in LRRK2 lead to increased mitochondrial fission and augmented mitochondrial reactive oxygen species (ROS) production, adversely affecting mitochondria and inducing the autosomal dominant form of PD [73].

### 2.3. Lysosomal Disorders

Lysosomal dysfunction triggers the onset of PD and the initial deposition of α-Syn lesions, accelerating the progression of the disease [74]. Damage to degradation activity and reduced proteostasis are features of aging, and lysosomes mediate the interaction between related genetic mutations and aging. This can influence the onset of PD [75,76]. Gradual abnormalities in endolysosomal function adversely affect other pathological mechanisms of PD, such as synaptic and mitochondrial dysfunctions.

#### 2.3.1. Autophagy–Lysosomal Pathway

Lysosomes play a crucial role in maintaining cellular homeostasis by regulating the metabolism of proteins, lipids, and large molecules through endocytosis, phagocytosis, and autophagy. Through the autophagy–lysosomal pathway, proteostasis of the α-Syn protein is maintained. However, lysosomal dysfunction hinders the removal of α-Syn, leading to its accumulation and propagation through synapses, causing cellular toxicity [77]. This forms a positive feedback loop where the toxic forms of α-Syn disrupt lysosomal biogenesis, contributing to the pathogenesis of PD.

Autophagy is the process by which cells eliminate abnormal or damaged proteins, damaged organelles, and unnecessary cellular components via lysosomes. Autophagy can be broadly classified into three types: macroautophagy, microautophagy, and chaperone-mediated autophagy (CMA). Macroautophagy involves the formation of autophagosomes, which fuse with lysosomes for content degradation. Microautophagy directly engulfs cytoplasmic components in the lysosome. CMA involves the direct transfer of substrate proteins through the lysosomal membrane, interacting with the lysosomal receptor lysosome-associated membrane protein 2a (LAMP2a) without forming additional vesicles [78,79]. PD patients exhibit abnormalities in these autophagy systems, with reduced levels of LAMP2a and heat shock cognate 71 kDa (Hsc70) protein in the CNS.

#### 2.3.2. Glucocerebrosidase (GCase)

Beta-glucosidase (GBA) is a gene encoding glucocerebrosidase, an enzyme that hydrolyzes glucosylceramide into ceramide and glucose [80]. Loss-of-function mutations in GBA are common and potent risk factors for PD. However, even among PD patients without GBA mutations, GCase activity can be reduced [81]. When GBA function is lost, accumulation of glucosylceramide and glucosylsphingosine due to GCase dysfunction disrupts autophagy, leading to α-Syn accumulation [82]. Accumulation of glucosylceramide promotes the formation and stabilization of oligomeric α-Syn intermediates in human-induced pluripotent stem cell-derived neuronal cultures [64]. Defects in lysosomal function can lead to lysosomal storage diseases, some of which can progress to neurodegeneration. It has been shown that patients with Gaucher disease and individuals with heterozygous mutations in GBA have an increased risk of developing PD. Loss-of-function mutations in the GBA1 gene, which encodes the lysosomal hydrolase β-glucocerebrosidase, lead to the accumulation of lipid substrates in Gaucher’s disease. This dysfunction is linked to Parkinson’s disease through the accumulation of α-synuclein in Lewy bodies and neurites, contributing to neuronal death. GBA1 mutations increase the risk of Parkinson’s disease, and Parkinson’s patients exhibit decreased β-glucocerebrosidase activity. Loss of β-glucocerebrosidase activity also promotes α-synuclein accumulation and toxicity, and conversely, α-synuclein accumulation further contributes to decreased lysosomal β-glucocerebrosidase activity by disrupting its trafficking to lysosomes. This disruption extends to other lysosomal hydrolases, exacerbating lysosomal dysfunction and neuronal dyshomeostasis [81,82].

#### 2.3.3. LRRK2

LRRK2 phosphorylates 14 members of the Rab family at the switch-II domain. Some of these Rab proteins promote vesicle transport from vesicles to the Golgi apparatus [83], thereby reducing cellular toxicity caused by α-Syn accumulation in PD models. Conversely, mutant LRRK2 delays early late and late endosomal trafficking. In PD patient fibroblasts, Rab7 activity was found to be reduced compared to healthy controls due to pathogenic LRRK2 mutations [64]. Additionally, mutant LRRK2 suppresses LAMP2a, inhibiting CMA. LRRK2 knockout mice showed abnormalities in protein removal functions, and increased cell death and oxidative stress were observed due to increased α-Syn [84].

In conclusion, mutant LRRK2 acts as a negative regulator of autophagy, leading to the accumulation of α-Syn in dopaminergic neurons [85]. Furthermore, α-Syn itself disrupts the normal function of the endosomal–lysosomal (E-L) system, further promoting α-Syn accumulation [86].

### 2.4. Inflammation

Patients with PD manifest neuroinflammation alongside the loss of dopaminergic neurons [87,88]. This is associated with alterations in microglia activity. When microglia are activated, Tumor necrosis factor alpha (TNF-α) releases various free radicals and activates numerous genes and proteins, including inducible nitric oxide synthase (iNOS) and inflammatory cytokines. The inflammatory molecule interleukin-1β (IL-1β) cytokine family stimulate Toll-like receptors (TLRs) and hinder the phagocytic action of microglia [89]. Endotoxins, such as lipopolysaccharides (LPS), also induce inflammatory responses through TLRs. When LPS binds with TLRs, it activates several signaling pathways, including mitogen-activated protein kinase (MAPK), phosphatidylinositol 3-kinase (PI3K)/protein kinase B (AKT), and the mechanistic target of rapamycin (mTOR), initiating the inflammation. This subsequently promotes nuclear factor-κB (NF-κB) activation, enhancing microglial phagocytic action and cytokine secretion. The NF-κB protein is activated extracellularly, but after interacting with inflammatory cytokines, it moves to other tissues causing damage [90].

#### 2.4.1. Microglia

There are two activation forms of microglia: M1 and M2. When the M1 phenotype is activated, it stimulates pathogen-associated molecular patterns (PAMPs), damage-associated molecular patterns (DAMPs), LPS, and amyloid-β aggregates, promoting dopaminergic neuronal damage. In PD models, M1 phenotype microglia are activated by both external and internal stimuli [91]. When the M2 phenotype is activated, it performs the opposite action to the M1 phenotype, producing various anti-inflammatory cytokines such as interleukins 4, 13, 10 (IL-4, IL-13, IL-10), glucocorticoids, and TGF-β [92,93]. The pathophysiology of inflammation in PD is significantly influenced by microglial activation and the polarization of microglia into M1 and M2 phenotypes, which play critical roles in neuroinflammation and neurodegeneration. In neurodegenerative diseases like PD, microglia-mediated neuroinflammation is a common feature. These cells can transition between different phenotypes, primarily classified into M1 and M2, although they also exhibit a range of intermediate states [94]. Microglia are at the forefront of the neuroinflammatory response in PD. Under normal conditions, they are in a “homeostatic” state, but pathological stimuli trigger a switch to a “reactive state.” The persistent activation of M1 microglia contributes to inflammatory diseases, including PD [95]. M1 microglia, when activated by pathogenic molecules like LPS, IFN, or protein aggregates (e.g., α-synuclein), release inflammatory molecules like ROS and pro-inflammatory cytokines (e.g., IL-1β, iNOS, TNF-α). This prolonged exposure to inflammatory mediators can result in neuronal damage. On the other hand, mediators like TGF-β, IL-4, IL-10, and IL-13 can induce the transition from M1 to M2. The M2 phenotype aids in phagocytosis, extracellular matrix (ECM) rebuilding, and neuronal survival, secreting factors such as Ym1 and FIZZ1 [95]. The phenotypic transformation of microglia from M1 to M2 is dynamic and associated with the stages and severity of neurodegenerative diseases. In later stages, M1 microglia predominate at the injury site, suppressing the immunoresolution and repair processes of M2 microglia. This transition is complex and is influenced by the presence of endogenous stimuli like aggregated α-synuclein, mutated superoxide dismutase, β-amyloid, and tau oligomers. These stimuli can continuously activate M1 pro-inflammatory responses, ultimately leading to irreversible neuron loss. Therefore, understanding and managing the stage-specific switching of microglial phenotypes could provide therapeutic benefits [96].

Research on the effects of natural and herbal medicine, specifically flavonoids, on the M2 anti-inflammatory response in neuroinflammation, including conditions like Parkinson’s Disease, has shown promising results. Flavonoids, natural compounds found in herbs, fruits, vegetables, tea, and cereals, have demonstrated an ability to alleviate neuroinflammation [97,98,99]. This effect is achieved by inhibiting the production of pro-inflammatory mediators and enhancing the secretion of anti-inflammatory factors. Flavonoids also modulate the polarization of microglia, shifting them from the M1 to the M2 phenotype. This modulation occurs through the suppression of the NLRP3 inflammasome, NF-κB, MAPK, and JAK/STAT pathways, and through the promotion of Nrf2, AMPK, BDNF/CREB, Wnt/β-Catenin, PI3k/Akt signals and SIRT1-mediated HMGB1 deacetylation [100]. Previous studies have focused on Shaoyao–Gancao Decoction (SGD), a traditional herbal medicine, which showed potential in promoting the M2 polarization of microglia. SGD induced upregulation of IL-13, which is known to drive microglia towards the M2 phenotype [101]. This transition was associated with anti-inflammatory and neuroprotective effects, suggesting that SGD could be a beneficial treatment in conditions involving neuroinflammation. Another study investigated the effects of luteolin, a flavonoid, on microglia polarization [102]. This study found that luteolin suppressed the production of pro-inflammatory mediators in LPS-stimulated primary microglia cells while promoting the expression of anti-inflammatory M2 markers. These effects were dose-dependent and were achieved by influencing the IL-17 and TNF signaling pathways, crucial for controlling microglia polarization. Specifically, luteolin was shown to inhibit the activation of the NF-κB pathway and reduce the expression of MMP9, a molecule involved in microglia activation and neuroinflammation. These results highlight luteolin’s potential as a therapeutic agent in managing neuroinflammation through modulation of microglia polarization.

#### 2.4.2. Astrocytes

Astrocytes are induced to secrete inflammatory cytokines in response to LPS. Excessive activation of astrocytes is recognized by morphological changes such as neuronal cell body hypertrophy and increased expression of astrocyte reactivity markers like glial fibrillary acidic protein (GFAP) [103]. When astrocytes are activated, NF-κB is activated, regulating the secretion of chemokines and adhesion molecules, assisting lymphocyte infiltration, and leading to neurodegeneration [104]. In PD, astrocytes, particularly their polarized A1 proinflammatory phenotype, play a crucial role in the progression of neuroinflammation and neurodegeneration. Astrocytes, derived from neural stem cells, are critical for maintaining the balance of ions, fluid, and transmitters in the brain, as well as regulating blood flow and the blood–brain barrier (BBB) [100]. In the context of neuroinflammation, such as in PD, astrocytes can polarize into a neurotoxic A1 phenotype. This state is characterized by cellular hypertrophy, increased production of glial fibrillary acidic proteins and complements, astrogliosis, glial scar formation, and the secretion of pro-inflammatory factors. These A1 astrocytes also affect vascular and perivascular cells, altering BBB permeability. The interaction between A1 astrocytes and microglia results in a feedback loop, exacerbating neuroinflammation and contributing to neuronal damage and excitotoxicity [100]. The morphological changes in astrocytes, particularly the transition to the A1 phenotype, are indicative of the stage and severity of PD. Preventing this transition, or reverting astrocytes to a less inflammatory state, has significant therapeutic potential. For example, GLP1R agonists have been suggested to inhibit the conversion of astrocytes to the A1 phenotype, offering neuroprotection in PD models. This inhibition is key in developing treatments for neurodegenerative diseases where no disease-modifying therapies currently exist [105].

The traditional Chinese herbal medicine Astragalus membranaceus Bunge and its derivative Astragaloside IV (AS-IV) have shown promising results. AS-IV exerts anti-inflammatory and neuroprotective activities, and is particularly effective against astrocyte senescence [106]. It has been observed to inhibit replicative and premature senescence in astrocytes, reducing the pro-inflammatory senescence-associated secretory phenotype [107,108]. This effect is mediated through the promotion of mitophagy, reducing damaged mitochondria accumulation and mitochondrial reactive oxygen species generation. Consequently, AS-IV protects against dopaminergic neuron loss and behavioral deficits in PD mouse models by reducing senescent astrocyte accumulation in the substantia nigra compacta [109].

#### 2.4.3. TLRs

TLRs play a crucial role as pattern recognition receptors (PRRs) in initiating immune responses to various stimuli [104,110]. In PD patients, TLRs identify α-Syn aggregates as DAMPs, playing a pivotal role in inducing and maintaining neuroinflammation. In PD patients, TLRs mediate inflammatory responses in peripheral blood leukocytes, glial cells, and neurons [89]. While TLR2, TLR4, and TLR9 are proven to have significant roles in PD pathogenesis, the relevance of other TLRs, such as TLR7 and TLR8, remains unclear [104]. In PD patient neural cell loss sites, TLR2 and CD68+ co-expressing microglia are activated [111]. The TLR2 receptor is responsible for microglial activation, releasing toxic factors that progress neural damage [89]. TLR4 is involved in microglial α-Syn phagocytosis, so a deficiency in TLR4 can impair the α-Syn clearance function, potentially causing neurodegeneration [111].

#### 2.4.4. Pro-Inflammatory Cytokines and Parkinson’s Disease

Cytokines are proteins acting as signaling molecules in the inflammation, infection, growth, and recovery of damaged tissues. They regulate the transportation of leukocytes in the brain and gather other proteins. Elevated concentrations of pro-inflammatory cytokines like TNF-α, IL-1β, and IL-6 have been observed in the substantia nigra pars compacta and cerebrospinal fluid of sporadic PD patients [112,113]. Such alterations in cytokine levels can be initiated by activated microglia, potentially promoting dopaminergic neuronal cell death [114]. Among them, IL-1β plays a significant role in neurodegenerative diseases like PD. It acts through IL-1 receptors found in neurons and microglia, leading to the expression of pro-inflammatory cytokines and perpetuating the inflammation cycle. In animal models of PD, administering IL-1 receptor antagonists significantly reduced TNF-α and interferon gamma (IFN-γ), decreasing the elevated loss of dopaminergic neurons induced by LPS [115].

## 3. Current Medications for Parkinson’s Disease

Commonly used treatments in current clinical settings aim to supplement dopamine in PD patients [116]. Medications such as Levodopa, designed to alleviate PD symptoms and improve quality of life, and others like the type-B monoamine oxidase (MAO-B) inhibitor, catechol-O-methyltransferase (COMT) inhibitor, and dopamine agonists have been developed and prescribed to address the drawbacks of Levodopa [117]. However, a complete cure for PD has not yet been discovered [118].

### 3.1. Levodopa

Levodopa is used to replenish dopamine in PD patients. It is a precursor of dopamine and, after crossing the blood–brain barrier (BBB), is decarboxylated by aromatic amino acid decarboxylase (AADC) to be converted into dopamine [119]. Dopamine transporters absorb exogenous dopamine from Levodopa, storing it in synaptic vesicles, maintaining a relatively stable dopamine level in the synaptic cleft during Levodopa treatment. Patients experience a stable response to Levodopa in the earlier stages of PD. However, improvements from Levodopa alone have limitations in treating Levodopa-induced dyskinesia [120]. Prolonged use of Levodopa might cause a reduction in its efficacy, and side effects such as motor complications (e.g., akinesia, bradykinesia) and psychological side effects (e.g., depression, cognitive impairment) can occur. Therefore, it is crucial to explore various potential treatments beyond dopamine supplementation using Levodopa [121].

### 3.2. COMT Inhibitor

COMT inhibitors are drugs that inhibit the enzyme COMT, which methylates substrates with a catechol moiety, limiting the availability of Levodopa in the brain by converting it to 3-O-methyl dopa [119]. Treating with COMT inhibitors prevents O-methylation of Levodopa and promotes its conversion to dopamine in the brain. However, the half-lives of the three widely used COMT inhibitors are very short, i.e., 2 to 3 h. This makes it challenging to achieve the desired inhibitory levels, often requiring nearly 10 doses daily, which can be inconvenient for patients. There are some adverse reactions, notably dopamine-related reactions like newly emerging dyskinesias, exacerbation of existing dyskinesias, nausea, and non-dopaminergic reactions such as diarrhea. Additionally, cases of acute liver failure have been reported, three of which resulted in death [122].

### 3.3. MAO-B Inhibitor

MAO catalyzes the oxidative deamination of monoamine neurotransmitters such as dopamine, phenylethylamine, 5-hydroxytryptamine, and norepinephrine. Among these, MAO-B metabolizes dopamine released in synapses and is absorbed by glial cells. MAO-B inhibitors block the metabolism of dopamine by inhibiting MAO-B activity, enhancing dopamine signaling, and selectively elevating dopamine levels in the synapse. They also offer various other benefits. The metabolism of dopamine by MAO-B forms toxic metabolites such as hydrogen peroxide and dihydroxyacetone. However, MAO-B inhibitors prevent this toxic process, thus protecting dopaminergic neurons in the substantia nigra and slowing the clinical progression of PD. They also inhibit nitric oxide synthase activity, reduce hydrogen peroxide production, improve brain mitochondrial function, and prevent the accumulation of α-Syn, thereby slowing the progression of PD [123]. However, Rasagiline, a well-known MAO-B inhibitor, showed no clear neuroprotective effects in clinical trials, limiting its therapeutic potential, and was found to have irreversible binding to active sites, causing long-term side effects. Selegiline, another MAO-B inhibitor, reported side effects such as insomnia, hallucinations, and akinesia. Additionally, safinamide, a reversible inhibitor with fewer side effects, has been reported to cause nausea, headaches, vomiting, fever, and high blood pressure [124,125].

### 3.4. Dopamine Agonist

Dopamine agonists provide significant anti-Parkinson symptom effects. They are used as the first-line drug for initial PD in patients under 60 years old, delaying the onset of motor complications and dysfunctions, and postponing the initiation of Levodopa therapy. Some authors advocate for starting PD treatment with dopamine agonists. However, dopamine agonist treatment is ineffective for patients who do not respond to Levodopa [126].

## 4. Evidence of Natural Products from Pre-Clinical Studies

Currently, drugs designed to treat PD only aim at alleviating the symptoms. However, there are no drugs that target the root cause of the disease. The most common treatment approach involves replenishing the dopamine levels in the brain. Yet, there are no therapies available that can repair the damaged brain cells. Especially in the realm of Western medicine, drugs are synthetic. The usual approach is to design a drug to act on one specific target pathway to minimize adverse side effects. However, there are some challenges where patients develop tolerance upon continued usage and increasing the dosage that leads to additional side effects. Given these challenges, there is a growing need for the development of new drugs. Our research explored alternative natural products to surpass the constraints of the current medicinal approach. Herbal medicines, made from plant-derived natural substances, have a diverse composition (Table 1). This means it is possible to address multiple pathways using a single drug, which may be a potential treatment avenue for PD (Figure 1).

### 4.1. Duzhong Fang

Duzhong Fang is a traditional Chinese medicine formula that consists of four ingredients: dried Eucommia ulmoides, Dendrobium, Rehmanniae Radix, and dried ginger. These ingredients are mixed in a specific weight ratio of 200:2:3:3 [127]. Duzhong Fang can regulate microglial morphology and reactivity, reducing microglial reactivity and inflammation in the central nervous system. Additionally, Duzhong Fang can directly inhibit the POMC gene, which is an upstream target for regulating inflammation and proinflammatory cytokines. By inhibiting POMC levels, Duzhong Fang can restore the homeostatic signature of microglia in Parkinsonian rats, leading to the alleviation of neuroinflammation and improvement of motor function. Duzhong Fang demonstrated anti-inflammatory effects in a mice model of PD, playing a role in managing neuroinflammation in PD by modulating microglial reactivity [128].

### 4.2. Kyung-Ok-Ko (KOK)

KOK’s original formula comprises juice from the root of Rehmannia glutinosa Liboschitz var. purpurae Makino (9.6 g), powder of dried fruit of Poria cocos Wolf (1.8 g), powder from the root of Panax ginseng C.A. Meyer (0.9 g), and honey (6 g). As a traditional composite herbal remedy, KOK has been employed to treat a wide range of diseases and conditions due to its action on multiple targets [129]. The MAPK (extracellular signal-regulated kinase ½ (ERK), c-Jun NH2-terminal kinase (JNK), and p38) and NF-κB signaling pathways, associated with neuronal loss, neuroinflammation, and BBB disruption in PD, can be potentially inhibited by KOK treatment according to the anti-inflammatory properties of KOK, similar to its role as a protective agent against MPTP-induced neurotoxicity [130].

### 4.3. Da-Bu-Yin-Wan (DBYW)

DBYW is composed of Amur corktree bark (Phellodendron chinense Cortex; Huang-Bai) 12 g, common Anemarrhena rhizome (Anemarrhenae Rhizoma; Zhi-Mu) 12 g, prepared rehmannia root (Radix Rehmanniae Praeparata; Shu-Di-Huang) 18 g, and tortoise shell (Carapax et Plastrum Testudinis; Gui-Jia) 18 g, as described in a prior study [131]. This study has shown that DBYW can elevate the expression of tyrosine hydroxylase (TH), enhance levels of monoamine neurotransmitters, and minimize mitochondrial DNA damage. The study demonstrates that DJ-1 overexpression increases Akt phosphorylation, leading to improved mitochondrial function and cell survival in a cellular model of Parkinson’s disease. Furthermore, DBYW was found to augment this process by enhancing the effects of DJ-1 on mitochondrial function through Akt phosphorylation. Additionally, DBYW enhances mitochondrial protection in PD by elevating cellular ATP content and decreasing the expression of ATP-sensitive potassium channel subunits [132].

### 4.4. Bee Venom Phospholipase A2 (BvPLa2)

BvPLa2 is an enzyme present in bee venom. This enzyme breaks down membrane phospholipids, producing free fatty acids and lysophospholipids. BvPla2 is known for its varied pharmacological effects, including anti-HIV activity, myotoxicity, and the promotion of neurite growth [133,134]. As a significant component of bee venom, BvPla2 stimulates regulatory T cells, mitigating neuroinflammatory reactions. This leads to the improvement of movement disorders and a reduction in α-Syn levels [135].

### 4.5. Hesperetin

Hesperetin is a flavonoid found in citrus fruits. It has shown protective effects against 6-hydroxydopamine (6-OHDA) lesions in rat striatum. Hesperetin treatment in 6-OHDA lesioned rats resulted in a reduction in apomorphine-induced rotational asymmetry, indicating an improvement in motor function. Additionally, hesperetin decreased the latency to initiate and the total time on the narrow beam task, suggesting an enhancement in motor coordination and balance. These effects indicate that hesperetin has the potential to improve motor function and coordination in rats with 6-hydroxydopamine-induced damage, possibly through its protective effects on the dopaminergic system and related motor pathways. It was found to provide protective effects in early models of PD by reducing behavioral disorders, alleviating oxidative stress, and reducing astrogliosis. Furthermore, hesperetin can be considered a potential adjunct therapy for PD management by preventing cell death and loss of dopaminergic neurons in the substantia nigra [136].

### 4.6. Paeonol

Paeonol is a major phenol compound from the Chinese herb Cortex Moutan, known for its antioxidant, anti-inflammatory, and anticancer properties [137]. This study showed that paeonol treatment significantly restored the activity of superoxide dismutase (SOD), catalase (CAT), and glutathione (GSH) in the midbrain, thereby alleviating oxidative stress induced by 1-methyl-4-phenyl-1,2,3,6-tetrahydropyridine/probenecid (MPTP/p). Additionally, Paeonol treatment decreased the levels of microglia and interleukin-1β (IL-1β), indicating a reduction in neuroinflammation, and increased the levels of brain-derived neurotrophic factor (BDNF), indicating neurotrophic effects on dopaminergic neurons. It has shown therapeutic effects against MPTP-induced PD in mice by improving behavioral tests, enhancing TH expression, reducing oxidative stress, and inhibiting microglial activation. Paeonol’s capability to promote neuronal survival and delay PD progression suggests its possibility as a treatment for PD [138].

### 4.7. Gastrodin

The Chinese herb Rhizoma Gastrodiae contains Gastrodin, a phenol glycoside known for its neuroprotective, antidepressant, and antiepileptic properties [139]. Gastrodin protects dopaminergic neurons, reduces the accumulation of α-Syn protein, and inhibits amyloid-β production and aggregation. In this study, Gastrodin rescued the climbing ability of Pink1 mutant flies, indicating an improvement in their motor function. Moreover, Gastrodin delayed the progressive loss of a cluster of dopaminergic neurons in the protocerebral posterial lateral 1 region of Pink1 mutant flies, and increased the dopamine content in the brain of Pink1 mutant flies. In other words, Gastrodin demonstrated potential anti-aging effects by extending lifespan, enhancing antioxidant capacity, and delaying PD-like phenotypes [140].

### 4.8. Trehalose

Trehalose is a natural disaccharide found in invertebrates, fungi, and many plants. It has value in various potential applications; has anti-inflammatory, antioxidant stress, and anti-cell death effects; and is a non-toxic compound [141,142]. In an AAV α-synuclein rat model of PD, animals that received unilateral AAV1/2 A53T α-synuclein showed a deficit in the use of their paw contralateral to the site of vector injection 3 weeks post-surgery, compared with empty vector controls. Neurochemical analysis demonstrated a significant attenuation in α-synuclein-mediated deficits in motor asymmetry and DA neurodegeneration, including impaired DA neuronal survival and DA turnover, as well as α-synuclein accumulation and aggregation in the nigrostriatal system, by commencing 5 and 2% trehalose at the same time as delivery of AAV. Trehalose protects against α-synuclein-mediated DA degeneration by enhancing autophagy in the striatum, which leads to the reduction of α-synuclein aggregates, improved DA neuronal survival, and prevention of behavioral asymmetry. Trehalose stimulates autophagy in an mTOR-independent manner, which helps in the clearance of α-synuclein aggregates [143].

### 4.9. Bu-Shen-Jie-Du-Fang (BSJDF)

BSJDF is a composite traditional Chinese medicine made up of Rehmannia glutinosa, Cistanche deserticola, Paeonia lactiflora Pall, Radix Angelica sinensis, Puerariae Radix, Rhizoma Coptidis, Radix Scutellariae, Antelope Horn Powder, and Glycyrrhizae Radix in a weight ratio of 5:5:4:4:5:4:4:1:2 [144,145]. BSJDF was found to protect PC12 cells by inducing autophagy in an MPP+-induced cell model of Parkinson’s Disease. The BSJDF group had the greatest surviving cell counts compared with all other treated cell groups except the normal group. Autophagy was observed in the BSJDF group by transmission electron microscopy (TEM), and protein expression of Atg12 and LC3 in the BSJDF group was upregulated compared to the PD model group. BSJDF was found to improve cell survival in an MPP+-induced cell model of Parkinson’s Disease by inducing autophagy, as evidenced by increased protein expression of Atg12 and LC3, and upregulated Atg12 mRNA expression. The study suggests that autophagy plays an important role in cell fate and maintaining cellular metabolic balance in Parkinson’s disease. These findings highlight the potential role of BSJDF in modulating autophagy and its implications for the development of treatments for Parkinson’s disease [146].

### 4.10. Nerolidol (NRD)

NRD is a sesquiterpene alcohol found in the essential oils of Baccharis dracunculifolia, Amaranthus retroflexus, and Canarium schweinfurthii. It exhibits various biological properties, including antioxidant and anti-inflammatory effects. NRD’s ability to easily cross the BBB enhances its potential as a treatment for neurodegenerative diseases like PD [147]. This study does not explicitly address whether NRD can be used as a standalone treatment for PD or if it needs to be combined with other medications. However, it is important to note that the neuroprotective effects of NRD against neuroinflammation and oxidative stress were demonstrated in an experimental model of PD induced by rotenone. NRD exerts its neuroprotective effects by reducing oxidative stress and neuroinflammation. It has been reported to increase the activities of antioxidant enzymes such as SOD and CAT, and to decrease the level of the antioxidant tripeptide GSH. Additionally, NRD inhibits the release of proinflammatory cytokines and inflammatory mediators, and prevents the activation of glial cells, ultimately attenuating neurodegeneration induced by rotenone. Furthermore, NRD has been shown to reduce the level of lipid peroxidation and nitrite content in the hippocampus, protecting against oxidative stress. NRD supplementation demonstrates promising neuroprotective effects by attenuating dopaminergic neurodegeneration, enhancing antioxidant enzyme activity, and inhibiting brain inflammatory mediators and lipid peroxidation [148].

### 4.11. Vanillic Acid (VA)

Vanillic acid, 4-hydroxy-3-methoxybenzoic acid, is the oxidized form of vanillin, and can be isolated from Gastrodia [149]. Co-treatment with VA and Levodopa–Carbidopa in a rotenone-induced Parkinson’s disease rat model showed significant effects on various Parkinson’s symptoms [150]. The co-treatment led to a significant reduction in muscle rigidity and catalepsy, along with a significant increase in body weight, rearing behavior, locomotion, and muscle activity in a dose-dependent manner, with the maximum effect observed at the 50 mg/kg dose of VA. Additionally, the co-treatment resulted in a significant increase in the level of dopamine in the VA plus standard drug-treated animals compared to the rotenone-treated group. Furthermore, histopathological evaluation showed a reduction in the number of eosinophilic lesions in the VA co-treated group compared to the rotenone group, indicating protection against neuronal damage due to oxidative stress and attenuation of motor defects. Furthermore, the study indicates a reduction in various oxidative stress markers in the brain, including increased lipid peroxidation and decreased GSH and catalase [149]. 

### 4.12. Vanillin

Vanillin, 4-hydroxy-3-methoxybenzaldehyde, an aromatic organic phenol molecule that can be extracted from Gastrodia, is widely used as a fragrance in the food, beverage, cosmetics, and pharmaceutical industries. This study demonstrated that Vanillin administration significantly increased striatal tissue dopamine content in 6-OHDA lesioned animals, which is a key indicator of nigrostriatal neurodegeneration. Additionally, Vanillin administration showed that attenuated 6-OHDA-induced rotations during apomorphine challenge, indicating its potential efficacy in reducing the phenotypic behavior associated with 6-OHDA lesion [151].

**Table 1 ijms-25-01071-t001:** Effect of natural products for PD.

Origin of Extraction	Mechanism	Cell or Animal Model	Inducer	Mode of Action and Target Signal	Site of Action (Figure 1)	Ref.
Duzhong Fang	Inflammation	C57bl/6 mice	MPTP	↓ locomotor dysfunction, inflammation, Iba1, microglia reactivity state ↑ striatal dopamine content, dopaminergic neurons, TH	3	[128]
KOK	Inflammation	C57BL/6 mice	MPTP ML385	↓ neurological dysfunction and motor impairments, the loss of dopaminergic neurons and fibers, Iba1, the upregulation of inflammatory mediators (IL-6, TNF-α, COX-2, and iNOS), neurotoxicity (microglial activation and inflammatory response ↓), BBB disruption markers (PECAM-1 and GFAP), neurotoxicity and inflammation (phosphorylated forms of ERK, JNK, and p38 & IκB and NF-κB ↓), ROS, MAPKs and NF-κB signaling pathways ↑ Nrf2 signaling (decreases the expression levels of Keap1 (a repressor protein that binds to Nrf2), and increases the expression levels of Nrf2 transcription factor, Nrf2 targeting genes HO-1 and NQO-1)	3	[130]
DBYW	Mitochondrial dysfunction	Rat PC-12 cells	pDJ-1 transfection MPP^+^	↓ DJ-1, mitochondrial dysfunction ↑ mitochondrial mass, total ATP content, the Akt phosphorylation	2	[132]
BvPLA2	Inflammation	Human A53T α-Syn Transgenic mice	A53T Transgenes	↓ motor dysfunction, α-Syn, the activation and numbers of microglia, and the ratio of M1/M2	3	[135]
Hesperetin	Inflammation	Wistar rats	6-OHDA	↓ astrogliosis (GFAP ↓), apoptosis (nigral DNA fragmentation ↓), the loss of SNC dopaminergic neurons ↑ striatal catalase activity and GSH content, Bcl2	3	[136]
Paeonol	Inflammation	C57BL/6 mice	MPTP	↓ motor dysfunction, oxidative stress (the activity levels of SOD, CAT, and GSH ↑), neuroinflammation(the number of Iba1-positive and IL-1β-positive cells ↓), ↑ TH-positive neurons, BDNF, dopaminergic neurons protection	3	[138]
Gastrodin	Mitochondrial dysfunction	Drosophila melanogaster	PINK1 gene mutant	↓ the loss of dopaminergic neurons, the onset of Parkinson-like phenotypes ↑ lifespan, climbing ability, resistance to oxidative stress, enzyme activities of superoxide dismutase (SOD) and catalase (CAT), the expression of anti-oxidative genes	2	[140]
Trehalose	Lysosomal Disorders	Human A53T α-Syn Transgenic mice	A53T Transgenes	↓ α-Synuclein-Induced Behavioral Impairment, α-Synuclein Accumulation ↑ DA Neuronal Survival, protection against the reduction of TH protein expression, autophagosome formation, LC3-II levels	1	[143]
BSJDF	Lysosomal Disorders	Pheochromocytoma12 (PC12)	MPP+ (MPTP)	improved cell survival in the PC12 cell PD model activated the autophagic process in PC12 cells. increased expression of Atg12 and LC3 proteins and upregulated Atg12 mRNA.	1	[146]
NRD	inflammation	Wistar rats	Rotenone	↑ level of superoxide dismutase, catalase, and glutathione ↓ level of malondialdehyde inhibited the release of proinflammatory cytokines and inflammatory mediators prevented ROT-induced glial cell activation and the loss of dopaminergic neurons and nerve fibers attenuated rotenone-induced dopaminergic neurodegeneration.	3	[148]
Vanillic acid	Mitochondrial dysfunction	Sprague Dawley rats	Rotenone	↓ Weight gain, Catalepsy, Rearing TBARS level (at 25 mg/kg and 50 mg/kg) SAG(superoxide anion generation) ↑ behaviour, CAT	2	[149]
Vanillin	Inflammation	Male Wistar rats	6-OHDA	↓ apomorphine-induced rotations, free radical release, expression of pro-inflammatory cytokines, lipid peroxidation ↑ striatal dopamine content, glutathione and superoxide dismutase enzyme protection of dopaminergic neurons	3	[151]

## 5. Evidence of Natural Products from Clinical Trials

Recently, clinical trials utilizing various natural substances for the improvement of PD symptoms have been conducted. These studies collectively suggest that natural products can potentially reduce neurotoxicity caused by oxidative stress and improve various symptoms associated with Parkinson’s disease.

Natural substances applied in the clinical trials include Licorice root, *Origanum majorana* L., and Pingchan. The trial results suggested that these natural substances are effective in reducing neurotoxicity caused by oxidative stress and improving motor symptoms. We, hereby, hypothesize that natural substances can be utilized in PD treatment [152].

According to Persian traditional literature [153], Licorice root was used for neurological conditions like headaches as a neuroprotective agent. In Persian folk medicine, formulations including Licorice extracts were used as neuroprotective herbal treatments for preventing disabilities associated with strokes or PD. In modern medicine, Licorice has shown various biological activities, such as anti-inflammatory, antioxidant [154], anti-tumor [155], antidepressant [156], memory-enhancing [157], neuroprotective [158], and anti-apoptotic effects. In a clinical trial involving 39 PD patients, patients were randomly divided into two groups and given either Licorice or a placebo syrup twice daily for six months. The results showed that adding Licorice extract as an adjunctive therapy can improve the overall Unified Parkinson’s Disease Rating Scale score of PD patients, improving daily activities, tremor and motor ability tests, and rigidity scores [152].

*Origanum majorana* L, known as sweet marjoram or simply marjoram, is an evergreen plant that belongs to the *Lauraceae* family. It is believed to have originated in the Mediterranean, North Africa, Egypt, and Asia and is widely used for culinary or medicinal purposes. In a clinical trial with 51 PD patients, patients who consumed *Origanum majorana* tea showed significant improvement in non-motor symptoms such as depression, anxiety, gastrointestinal and urinary symptoms, restlessness, and fatigue compared to the placebo group. Furthermore, consuming *Origanum majorana* tea daily for a month did not have adverse effects on liver and kidney functions. Notably, *Origanum majorana* L is rich in antioxidants like polyphenols and monoterpenes. Hence, this plant can neutralize reactive oxygen species and delay neurodegeneration [159].

Pingchan granule, a traditional Chinese herbal formulation, is composed of nine widely recognized herbs. These include *Lycium barbarum* L., *Taxillus chinensis* (DC.) Danser, *Gastrodia elata* Blume, *Paeonia lactiflora* Pall., *Arisaema erubescens* (Wall.) Schott, and *Curcuma phaeocaulis* Valeton. Additionally, it incorporates Bombyx mori Linnaeus, *Buthus martensii* Karsch, and *Scolopendra subspinipes mutilans* L. Koch. Each of these herbs has been carefully selected for their unique medicinal properties and synergistic effects in this composite herbal remedy. A randomized double-blind placebo-controlled trial was conducted on a cohort of 292 mild-to-moderate PD patients across multiple centers. This trial demonstrated the superiority of Pingchan granule over the placebo in managing both motor and non-motor symptoms of PD over a treatment period of 24 weeks. This effect persisted during the 12-week follow-up period. The Pingchan granule group showed better improvements in motor outcomes at timepoints 1, 2, and 3 compared to the placebo group, and demonstrated efficacy across a range of both motor and non-motor symptoms, suggesting the long-term beneficial effects of Pingchan granule on both motor and non-motor symptoms [160] of PD. The results of these trials can provide as strong evidence that the development of natural product-based medicines would be expected to have a very positive effect on PD treatment. However, further research with larger sample sizes and longer study durations is necessary to fully understand their long-term effects and potential in PD treatment.

## 6. Discussion

PD is a neurodegenerative disorder resulting from the progressive degeneration of dopamine-producing neurons in a brain region called the substantia nigra [161]. A primary pathology of PD is the widespread distribution of abnormal aggregates of the protein α-Syn. [162] This abnormal protein accumulation induces mitochondrial dysfunction, lysosomal disorders, and neuroinflammation, accelerating the death of dopamine-producing neurons [163,164,165]. Currently, medications such as Levodopa, COMT inhibitors, MAO-B inhibitors, and dopamine agonists are used to treat patients with this condition. However, these drugs only slow down the progression of the disease and do not cure it. They fail to fully address various non-motor symptoms like depression and sleep disorders and, thus, can cause various side effects [166,167]. However, various natural products such as curcumin, epigallocatechin gallate (EGCG), resveratrol, and quercetin have demonstrated minimal or negligible side effects, even at higher oral doses, in both animal and human studies [168]. These safety profiles are a primary reason that natural products are increasingly utilized in drug development. These products exhibit considerable advantages in terms of safety, underlining their potential as therapeutic agents in various medical applications.

Currently, due to the inefficacy, resistance issues, and side effects of single synthetic drugs, there is a shift from chemical monotherapy to multi-drug therapy. Multiple mechanisms and genes can be involved in a single disease, requiring multi-drug strategies [169]. Plant extracts contain various bioactive compounds that can interact with multiple targets in the body to produce therapeutic effects. When the multiple compounds in plant extracts interact with different targets, a synergistic effect can occur, enhancing the overall pharmacological impact [170]. Herbal medicine works in a similar manner, with complex interactions among various bioactive ingredients in herbs and herbal formulations leading to synergistic therapeutic effects. Moreover, synergistic interactions between herbal medicine and pharmaceuticals can enhance therapeutic effects or reduce side effects [171]. Therefore, using plant extracts and herbal medicines as therapeutic agents can complement the limitations of current treatments (e.g., levodopa) by acting on pathways not targeted by them.

Clinical trials utilizing herbal medicines and natural products to alleviate PD symptoms demonstrate that these materials exhibit various biological activities and can be extensively applied to target both motor and non-motor symptoms of PD. This suggests that herbal medicines and natural products can be employed in PD treatment, reducing neurotoxicity and improving both motor and non-motor symptoms [152,159,160].

There are limitations when developing drugs based on natural products. Due to the lack of standardized procedures for producing natural products, the quality and composition of natural products intended for medicinal use can vary greatly. Likewise, there is a scarcity of clinical trials conducted under standardized criteria, which subsequently results in insufficient scientific evidence for the therapeutic use of natural products. Thus, more clinical trials and research are required to verify the safety of these natural-product-based treatments in the human body under standardized criteria. Furthermore, because natural products contain a mix of various chemicals acting simultaneously, it is challenging to identify and isolate the active ingredients responsible for specific therapeutic effects. Moreover, research on the biological mechanisms through which these individual components act in the body is limited, calling for further investigation [172].

## Figures and Tables

**Figure 1 ijms-25-01071-f001:**
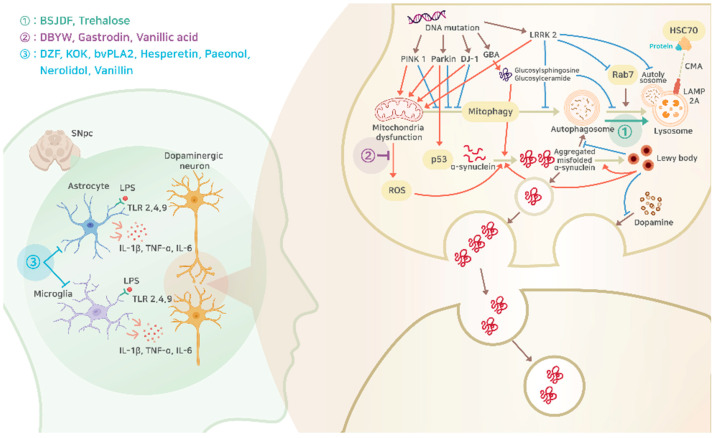
Schematic pathophysiologic mechanism of PD with potential intervention of natural products, the details of which are explained in the following sections. Circled number 1 (green letters), promotion of α-Syn removal through autophagy; circled number 2 (purple letters), inhibition of ROS synthesis from mitochondria dysfunction; circled number 3 (blue letters), inhibition of inflammatory responses from astrocyte and microglia. BSJDF, Bu-Shen-Jie-Du-Fang; DBYW, Da-Bu-Yin-Wan; DZF, Duzhong Fang; KOK, Kyung-Ok-Ko; bvPLA2, bee venom phospholipase A2; SNpc, substantia nigra pars compacta; LPS, lipopolysaccharide; TLR, toll-like receptor; IL, interleukin; TNF-α, tumor necrosis factor α; LRRK2, leucine-rich repeat serine/threonine-protein kinase 2; HSC70, heat shock cognate 71 kDa; ROS, reactive oxygen species; CMA, chaperone-mediated autophagy; LAMP2A, lysosome-associated membrane protein 2a.

## Data Availability

Not applicable.

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
