# Peer review of "The Potentiality of Natural Products and Herbal Medicine as Novel Medications for Parkinson’s Disease: A Promising Therapeutic Approach"

_ijms, 2024, doi:10.3390/ijms25021071_

Round 1

Reviewer 1 Report

Comments and Suggestions for Authors

This review article is titled "The potentiality of Natural Products and herbal medicine as Novel Medications for Parkinson's Disease: A Promising Therapeutic Approach." The author's natural products and herbal medicines have been a matter of curiosity in Parkinson's disease (PD) management due to their potential neuroprotective and symptomatic relief properties. However, it's crucial to approach their use cautiously and under healthcare professionals' direction, as their efficacy and safety might vary, and interactions with established medications could occur.

I have a few minor questions.

In line 247, the author discussed microglia. Could you include any specific M1 proinflammatory altered markers during the early and late stages of disease progression with the effect of natural and/or herbal medicine?

Is there any change in microglia cell morphology?

Discuss also M2-antiinflammatory changes with the effect of natural and/or herbal medicine?

In line 255, the author discussed astrocytes. Could you include any specific A1 proinflammatory altered markers during the early and late stages of disease progression with the effect of natural and/or herbal medicine?

Is there any change in microglia cell morphology?

Discuss also A2-antiinflammatory changes with the effect of natural and/or herbal medicine?

The authors very systematically covered related information about the title of the paper.

Why do you want to present a general review paper instead of a specific signaling or, gene or protein effect of natural or herbal medicine?

Reviewer 2 Report

Comments and Suggestions for Authors

This review has much value provided that it makes a stronger research into the evidence of the specific cellular models and the pre and post medical trials regarding the mentioned active compounds and natural products. However, the balance regarding PD and actual evidence of these compounds from the overall manuscript is tipping towards mostly explaining PD and cellular pathways. Sections 1 through 3, are much longer and detailed than section 4+5, the actual purpose of the paper. This needs to be rebalanced, not by reducing 1-3, but by expanding 4 and 5. See the specific suggestions below.

The only Figure needs to be improved and enhanced, as it given to be the overview of pathways and where the active natural compounds can influence PD related dysfunctions. For the included table, which needs is fine as an overview, but its text needs to be much more integrated and explained in section 4.

Many statements, especially in beginning of multiple sections/paragraphs, are missing. And often a citation at the end of a paragraph is linked to a manuscript-specific sub-conclusion, which is improper.

Section 5 on natural products does not seem to be properly linked to the overall manuscript, and it is required to investigate whether the single mentioned active (antioxidant) compounds mentioned under section 4, are available in Licorice root and/or marjoram and/or pingchan granule? Adding, additional writing on curcumin, epigallocatechin gallate (EGCG), resveratrol, and quercetin and their relevance to reducing PD symptoms might give the paper much more body.

Abstract

Lines 13-22 make up 2/3 of the abstract and is basically introduction on PD, dementia and associated cellular disfunction. This could be densed down.

Line 23 should be rephrased: ‘Thus, there is a demand for treatments with fewer side effects, with much potential offered by natural products.’

Lines 24/30 should be rephrased: ‘In this study, we reviewed a total of 14 articles related to herbal medicines and natural products and investigated their relevance to possible PD treatment. The results showed  that the reviewed herbal medicines and natural products are effective against lysosomal disorder, mitochondrial dysfunction, and inflammation, key mechanisms underlying PD. Therefore, natural products and herbal medicines can reduce neurotoxicity and might improve both motor and non-motor symptoms associated with PD. Furthermore, these products, with their multi-target effects, enhance bioavailability, inhibit antibiotic resistance, and might additionally eliminate side effects, making them good alternative therapies for PD treatment.’

1.      Introduction, should be given at Line 34.

Line 60: give multiple citations, not just one.

Line 63 give references for non-motor symptoms of PD.

Line 71 give references for falls and central features.

Line 77 give multiple references for PD drugs side effects.

Line 80 give multiple references for the surge of Korean traditional medicine and natural extracts.

Line 82, please use Beal et al., 2014 somewhere else, and not as the objective if your review.

2.      Phathophysiology

Lines 84/88 should have references.

Line 92/93 should read ‘Figure 1. Schematic pathophysiologic mechanism of PD with potential intervention of natural products of which the details are explained in the following sections Additionally, all abbreviations should be listed in the subscript (IL, TNF, etc.), and the figure needs to be of higher resolution with larger fonts. Generally, the figure should be included as a larger format, especially since it is the only figure that is supposed to contain all the pathways, drugs and compounds to be referred to in the following paragraphs.

2.1   Main cause

Line 120, please cite the Braak model.

Line 140, give multiple references for the MRI studies.

Line 142 give reference for alpha-syn in skin and intestines.

2.2   Mitochondrial dysfunction

Line 153/156 should read and needs referenced ‘have been associated with PD-like symptoms, have been shown to inhibit mitochondrial complexes (references), and the resulting disturbances in energy production and cellular functions could lead to damage in dopaminergic neurons and symptoms similar to PD.’

Line 157 needs references.

Lines 158-159 need references.

2.3   Lysosomal disorders

‘A few genes have been confirmed to cause familial PD,’ line 166 need references.

Line 184/186 needs references.

Line 217 explain how Gaucher disease pathology connects to lysosomal malfunction.

Line 233/234 need references.

Lines 263/265 need references.

Lines 275/278 need references.

3. Current medications

Line 317 should be changed to ‘3.3. MAO-B Inhibitor’

4. Evidence of natural products

Lines 357/359 need references

Lines 363/365 need references.

Lines 365/367 need references.

Lines 370 should read ‘…can be potentially inhibited by KOK treatment according to anti-inflammatory properties of KOK, similar to its role as protective agent against MPTP-induced neurotoxicity [79].’ Omitting lines 372/373 as closing sentence.

Line 378 ‘as described in prior studies’ should cite these references.

Line 378 and further, starts with ‘studies’, but only mentions one (80).

Lines 285/386 mentions ‘BvPla2 is known for its varied pharmacological effects,’ however no citations are given.

Lines 398/399 need references.

Lines 405/408 need references.

Lines 412/414 need references.

Table 1 Duzhong Fang reference 78 is mentioned twice. The content of this table needs to be more integrated and explained in the text, as these are the only observations relevant for the content of this review. In principle, the text is now loosely connected to the table, and does not make a solid enough plea for the use of the traditional natural compounds against PD symptoms.

Lines 418/421 need references.

Lines 421/423 need strong references, especially since ‘clinical observations’ and ‘studies’ is used.

Lines 427/430 need references, especially for the compound crossing the BBB.

Lines 435/439 need references. Especially also the PC12 cell study!

Lines 443, the rat glioma C6 study and the fruit fly model all need to be cited properly.

Lines 452/3 need references.

Line 454 the rat model is probably reference 89, but needs to be mentioned earlier.

5. Evidence

Lines 469/470 needs to be references by 95, as the citation at the end of line 474 seems improper.

Give the references for ‘According to Persian traditional literature’ in Line 475.

Give references for ‘In modern medicine’ line 478 and beyond.

Are any of the active (antioxidant) compounds mentioned under section 4, available in Licorice root and/or marjoram and/or pingchan granule? This needs to be properly explored and written down, otherwise these observations (and this section) come out of nowhere and are unlinked to the content of the paper. Interestingly, lines 518/519 mention curcumin, epigallocatechin gallate (EGCG), resveratrol, and quercetin, but have not been added to this section, as they could certainly give more body to the whole paper.

Line 489 the clinical trail refers to 96?, the reference at the end of the paragraph (line 495), seems improper.

Please reformulate lines 496/499 and give references.

Line 499/500 the trail probably refers to 97?, the reference at the end of the paragraph (line 507), seems improper.

Discussion

Line 528 ‘like’ should be replaced with ‘similar’?

Comments on the Quality of English Language

This review has much value provided that it makes a stronger research into the evidence of the specific cellular models and the pre and post medical trials regarding the mentioned active compounds and natural products. However, the balance regarding PD and actual evidence of these compounds from the overall manuscript is tipping towards mostly explaining PD and cellular pathways. Sections 1 through 3, are much longer and detailed than section 4+5, the actual purpose of the paper. This needs to be rebalanced, not by reducing 1-3, but by expanding 4 and 5. See the specific suggestions below.

The only Figure needs to be improved and enhanced, as it given to be the overview of pathways and where the active natural compounds can influence PD related dysfunctions. For the included table, which needs is fine as an overview, but its text needs to be much more integrated and explained in section 4.

Many statements, especially in beginning of multiple sections/paragraphs, are missing. And often a citation at the end of a paragraph is linked to a manuscript-specific sub-conclusion, which is improper.

Section 5 on natural products does not seem to be properly linked to the overall manuscript, and it is required to investigate whether the single mentioned active (antioxidant) compounds mentioned under section 4, are available in Licorice root and/or marjoram and/or pingchan granule? Adding, additional writing on curcumin, epigallocatechin gallate (EGCG), resveratrol, and quercetin and their relevance to reducing PD symptoms might give the paper much more body.

Abstract

Lines 13-22 make up 2/3 of the abstract and is basically introduction on PD, dementia and associated cellular disfunction. This could be densed down.

Line 23 should be rephrased: ‘Thus, there is a demand for treatments with fewer side effects, with much potential offered by natural products.’

Lines 24/30 should be rephrased: ‘In this study, we reviewed a total of 14 articles related to herbal medicines and natural products and investigated their relevance to possible PD treatment. The results showed  that the reviewed herbal medicines and natural products are effective against lysosomal disorder, mitochondrial dysfunction, and inflammation, key mechanisms underlying PD. Therefore, natural products and herbal medicines can reduce neurotoxicity and might improve both motor and non-motor symptoms associated with PD. Furthermore, these products, with their multi-target effects, enhance bioavailability, inhibit antibiotic resistance, and might additionally eliminate side effects, making them good alternative therapies for PD treatment.’

1.      Introduction, should be given at Line 34.

Line 60: give multiple citations, not just one.

Line 63 give references for non-motor symptoms of PD.

Line 71 give references for falls and central features.

Line 77 give multiple references for PD drugs side effects.

Line 80 give multiple references for the surge of Korean traditional medicine and natural extracts.

Line 82, please use Beal et al., 2014 somewhere else, and not as the objective if your review.

2.      Phathophysiology

Lines 84/88 should have references.

Line 92/93 should read ‘Figure 1. Schematic pathophysiologic mechanism of PD with potential intervention of natural products of which the details are explained in the following sections Additionally, all abbreviations should be listed in the subscript (IL, TNF, etc.), and the figure needs to be of higher resolution with larger fonts. Generally, the figure should be included as a larger format, especially since it is the only figure that is supposed to contain all the pathways, drugs and compounds to be referred to in the following paragraphs.

2.1   Main cause

Line 120, please cite the Braak model.

Line 140, give multiple references for the MRI studies.

Line 142 give reference for alpha-syn in skin and intestines.

2.2   Mitochondrial dysfunction

Line 153/156 should read and needs referenced ‘have been associated with PD-like symptoms, have been shown to inhibit mitochondrial complexes (references), and the resulting disturbances in energy production and cellular functions could lead to damage in dopaminergic neurons and symptoms similar to PD.’

Line 157 needs references.

Lines 158-159 need references.

2.3   Lysosomal disorders

‘A few genes have been confirmed to cause familial PD,’ line 166 need references.

Line 184/186 needs references.

Line 217 explain how Gaucher disease pathology connects to lysosomal malfunction.

Line 233/234 need references.

Lines 263/265 need references.

Lines 275/278 need references.

3. Current medications

Line 317 should be changed to ‘3.3. MAO-B Inhibitor’

4. Evidence of natural products

Lines 357/359 need references

Lines 363/365 need references.

Lines 365/367 need references.

Lines 370 should read ‘…can be potentially inhibited by KOK treatment according to anti-inflammatory properties of KOK, similar to its role as protective agent against MPTP-induced neurotoxicity [79].’ Omitting lines 372/373 as closing sentence.

Line 378 ‘as described in prior studies’ should cite these references.

Line 378 and further, starts with ‘studies’, but only mentions one (80).

Lines 285/386 mentions ‘BvPla2 is known for its varied pharmacological effects,’ however no citations are given.

Lines 398/399 need references.

Lines 405/408 need references.

Lines 412/414 need references.

Table 1 Duzhong Fang reference 78 is mentioned twice. The content of this table needs to be more integrated and explained in the text, as these are the only observations relevant for the content of this review. In principle, the text is now loosely connected to the table, and does not make a solid enough plea for the use of the traditional natural compounds against PD symptoms.

Lines 418/421 need references.

Lines 421/423 need strong references, especially since ‘clinical observations’ and ‘studies’ is used.

Lines 427/430 need references, especially for the compound crossing the BBB.

Lines 435/439 need references. Especially also the PC12 cell study!

Lines 443, the rat glioma C6 study and the fruit fly model all need to be cited properly.

Lines 452/3 need references.

Line 454 the rat model is probably reference 89, but needs to be mentioned earlier.

5. Evidence

Lines 469/470 needs to be references by 95, as the citation at the end of line 474 seems improper.

Give the references for ‘According to Persian traditional literature’ in Line 475.

Give references for ‘In modern medicine’ line 478 and beyond.

Are any of the active (antioxidant) compounds mentioned under section 4, available in Licorice root and/or marjoram and/or pingchan granule? This needs to be properly explored and written down, otherwise these observations (and this section) come out of nowhere and are unlinked to the content of the paper. Interestingly, lines 518/519 mention curcumin, epigallocatechin gallate (EGCG), resveratrol, and quercetin, but have not been added to this section, as they could certainly give more body to the whole paper.

Line 489 the clinical trail refers to 96?, the reference at the end of the paragraph (line 495), seems improper.

Please reformulate lines 496/499 and give references.

Line 499/500 the trail probably refers to 97?, the reference at the end of the paragraph (line 507), seems improper.

Discussion

Line 528 ‘like’ should be replaced with ‘similar’?

Round 2

Reviewer 2 Report

Comments and Suggestions for Authors

The manuscript has been improved substantially, especially by adding (and correctly re-positioning) citations, and elaborating on the PD pathophysiology regarding microglia and astrocytes, and editing the natural product section. Still the manuscript needs additional revision before acceptance and below are issues to resolve for further improvement.

Lines 297-300 needs references, especially ‘Flavonoids, ….alleviate neuroinflammation’.

Line 307 should cite 98. Do not place the citation at the end of line 310

Line 311, use the citation for ‘another study’. Probably it is 99, but the citation at the end of line 319 is not fully right.

Lines 345/346 should have citations.

Lines 348/349 ‘It has been observed….secretory phenotype’ needs a citation.

Figure 1 could be larger. Does MDPI not have a policy to use page space available, not only have the figure column-text wide?

It is strongly recommended to integrate Figure 1 and the text more clearly by doing the following:

-        In the subtext of the Figure shortly describe the three different coloured intervention sites.

-        Refer to the intervention sites of Figure 1 in the text of section 2.4.1 and 2.4.2, as well as throughout the manuscript were appropriate.

-        In table 1, refer to the three different coloured intervention sites for each specific compound.

Each natural product of section 4 should have a citation for its use, active compounds or composition, and then additionally for its effects. They might be the same study, but at least these aspects need to be referenced properly.

Line 635 Pingchan granule, a traditional Chinese herbal formulation, is composed of nine widely recognized herbs’ should have a reference.

Line 648/649, please citation 154 directly after a range of both motor and non-motor symptoms [154],

Line 650 the results of these trials. Which trials? Use citations.
